# Rapid and non-invasive detection of malaria parasites using near-infrared spectroscopy and machine learning

**Maggy T. Sikulu-Lord**[1]*, **Michael D. Edstein**[2], **Brendon Goh**[1], Anton R. Lord[3], Jye A. Travis[2], Floyd E. Dowell[4], Geoffrey W. Birrell[2], Marina Chavchich[2]

1 School of the Environment, Faculty of Science, The University of Queensland, Brisbane, Queensland, Australia, 2 Department of Drug Evaluation, Australian Defence Force Malaria and Infectious Disease Institute, Brisbane, Queensland, Australia, 3 Centre for Data Science, Queensland University of Technology, Brisbane, Queensland, Australia, 4 Center for Grain and Animal Health Research, USDA Agricultural Research Service, Manhattan, Kansas, United States of America

* maggy.lord@uq.edu.au

## Abstract

### Background

Novel and highly sensitive point-of-care malaria diagnostic and surveillance tools that are rapid and affordable are urgently needed to support malaria control and elimination.

### Methods

We demonstrated the potential of near-infrared spectroscopy (NIRS) technique to detect malaria parasites both, *in vitro*, using dilutions of infected red blood cells obtained from *Plasmodium falciparum* cultures and *in vivo*, in mice infected with *P. berghei* using blood spotted on slides and non-invasively, by simply scanning various body areas (e.g., feet, groin and ears). The spectra were analysed using machine learning to develop predictive models for infection.

### Findings

Using NIRS spectra of *in vitro* cultures and machine learning algorithms, we successfully detected low densities ($<10^{-7}$ parasites/µL) of *P. falciparum* parasites with a sensitivity of 96% (n = 1041), a specificity of 93% (n = 130) and an accuracy of 96% (n = 1171) and differentiated ring, trophozoite and schizont stages with an accuracy of 98% (n = 820). Furthermore, when the feet of mice infected with *P. berghei* with parasitaemia $\geq$3% were scanned non-invasively, the sensitivity and specificity of NIRS were 94% (n = 66) and 86% (n = 342), respectively.

### Interpretation

These data highlights the potential of NIRS technique as rapid, non-invasive and affordable tool for surveillance of malaria cases. Further work to determine the potential of NIRS to

**Data Availability Statement:** The minimum data set necessary to replicate our study's findings can be accessed through the OSF repository (Link: https://osf.io/u2c4f).

**Funding:** M.S-L., M.D.E. A.R.L., G.W.B., F.E.D., M.
C; APP 1159384; National Health and Medical
Research Council, Australia; https://www.nhmrc.
gov.au/funding. M.S-L.; AQIRF019-2018; Advance
Queensland Industry Research Fellowship; https://
advance.qld.gov.au/grants. Australian Defence
Organisation. The funders had no role in study
design, data collection and analysis, decision to
publish, or preparation of the manuscript.

**Competing interests:** The authors have declared
that no competing interests exist.

detect malaria in symptomatic and asymptomatic malaria cases in the field is recommended
including its capacity to guide current malaria elimination strategies.

## Introduction

With nearly half of the world's population at risk, malaria remains the deadliest mosquito-
borne disease in the world. In 2021, an estimated 619,000 deaths and 247,000 million malaria
cases were reported in 84 countries with almost 95% of global cases occurring in the WHO
African region mostly among children under the age of 5 years [1]. Over the last two decades,
efforts to reduce malaria transmission including the implementation of innovative vector con-
trol interventions and highly effective drug therapies have led to a significant reduction in
malaria cases especially in Southeast Asia prompting many countries to commit to malaria
elimination by 2030 [2]. However, since 2015, these achievements have reportedly stalled. This
is largely driven by a reservoir of undiagnosed asymptomatic malaria carriers with submicro-
scopic infections which can be as high as 60–90% in low transmission settings [3–9]. It is esti-
mated that asymptomatic malaria carriers are responsible for 45–75% of malaria transmission.
To facilitate the elimination of malaria, more sensitive, faster and cheaper diagnostic tools to
detect and facilitate treatment of asymptomatic carriers with sub-microscopic parasitaemia are
required.

For over a century, the diagnosis of malaria depended on the availability of medical facilities
and trained personnel using microscopy and Giemsa, Wright's, or Field's stained blood smears
[10]. Microscopy remains the preferred technique for malaria diagnosis. With a limit of detec-
tion of 50–100 parasites/μL [11, 12], it can also differentiate and quantify parasite species.
However, accurate diagnosis by microscopy is dependent on the quality of slides and availabil-
ity of expert microscopists with the ability to differentiate and quantify multiple species using
well prepared slides [13]. Three decades ago, rapid diagnostic tests (RDTs) based on Histidine-
rich protein II (HRP$_2$) antigen coupled with *Plasmodium* pan specific lactate dehydrogenase
(pLDH) or p-aldolase were developed [11]. RDTs are now widely utilised for malaria diagnosis
because they are easy to use and do not require qualified personnel. However, like microscopy,
they can only reliably detect >100 *P. falciparum* parasites/μL [11]. Furthermore, RDTs can
produce false positives due to persistence of HRP2 antigens in blood days post infection clear-
ance [14] and false negatives if parasites harbour HRP 2/3 deletions [15], as well as having diffi-
culties differentiating *Plasmodium species* [11, 16]. Recently, novel point of care device, Gazelle
(Hemex Health™), based on detection of malaria pigment, hemazoin, has been tested in several
countries achieving sensitivity similar to those of microscopy and RDTs [17] for vivax but less
sensitive for detecting falciparum malaria, although higher sensitivities have been reported for
the detection of asymptomatic infections [18].

Nucleic acid (DNA and RNA) amplification-based methods such as polymerase chain reac-
tion (PCR) including ultrasensitive quantitative PCR [19], real time quantitative PCR (qPCR),
reverse-transcriptase quantitative PCR (RT-qPCR) [20, 21], nested PCR [22–24] and loop-
mediated isothermal amplification (LAMP) of DNA or RNA [25], are the most sensitive and
most specific diagnostic tools for detecting submicroscopic malaria. Although these methods
have a limit of detection of 0.002–0.02 parasites/μL [26], they are expensive, technically
demanding, time consuming and mainly restricted to the laboratory settings. With no alterna-
tive currently available, qPCR methods have been used for the detection of asymptomatic
infections to support countries elimination efforts.

Blood film microscopy, RDTs and PCR-based methods are also invasive techniques requiring a finger blood capillary prick or even a higher volume of venous blood sample, which can reduce the acceptance of testing in asymptomatic populations. To achieve malaria elimination, more sensitive, point-of-care, fast and affordable detection tools are needed to facilitate detection and prompt treatment of asymptomatic carriers with sub-microscopic parasitaemia. The development of an affordable, rapid and non-invasive diagnostic tool would be a game changer in the battle towards worldwide malaria elimination.

The near-infrared spectroscopy (NIRS) technique uses the near-infrared region of the electromagnetic spectrum (800–2500 nm). When NIR light is shone on biological samples, some of the light is absorbed by the sample and some is reflected creating a spectral signature of the sample. The spectral signature corresponds to combinations and overtones of vibrational frequencies of chemical bonds of the molecules present in the sample. Machine learning algorithms are trained using spectral signatures with diagnostic features specific to the chemical profile of a sample and the resultant models are used to predict the identity of unknown samples. NIRS is a rapid, non-invasive, reagent-free technology able to detect and quantify specific components in biological samples, thus it can be potentially used for screening large populations such as the asymptomatic malaria populations.

NIRS has been applied to non-invasively detect various pathogens in mosquito samples including *Wolbachia* [27, 28], malaria [29], Zika [28, 30], Chikungunya [28] and to characterize the identity and age of *Anopheles* [31, 32] and *Aedes* mosquito vectors [33, 34]. Previously a related technique with longer wavelengths (2500–25000 nm) that uses the mid-infrared (MIR) region of the electromagnetic region was successfully demonstrated in the laboratory [35] and in the field as a potential diagnostic tool for malaria [36, 37]. However, a limitation of the MIR technique is the inability to analyse samples non-invasively. More recently a miniaturized NIRS scanner was used to detect *Plasmodium falciparum* parasites *in vitro* [38] and non-invasively through the skin of humans subjects [39].

Here, we demonstrate the sensitivity of NIRS to detect low parasite densities and asexual stages of *P. falciparum in vitro* and its capacity to non-invasively detect malaria parasites in mice infected with *P. berghei* at varying parasitaemia levels. We also compared the sensitivity of NIRS when used non-invasively against its sensitivity in blood samples collected from *P. berghei* infected and uninfected mice.

## Methods

### Cultivation of *P. falciparum* rings, trophozoites and schizonts including the preparation of blood slides for detection by NIRS

"Base" medium was prepared by combining 10.43 g/L RPMI 1640, 5.95 g L HEPES, 2 g/L D-glucose, 50 mg/L hypoxanthine, with pH adjusted to 7.0, followed by sterilising by filtering through 0.22 μM Millex® filter (Millipore, USA). "Plain" medium, used for washing, was prepared immediately before use by supplementing the "base" medium with $NaHCO_3$ to 0.22%. "Complete" medium was made by adding the heat-inactivated (56°C for 1 h) human plasma to 10% (vol/vol) to the "plain" medium and was used to culture parasites as previously described [47]. Briefly, *P. falciparum* D6 (Sierra-Leone) or 3D7 (Netherlands) laboratory lines were cultured in "complete" medium containing 4% of $O^{(+)}$ RBC (LiveBlood, Brisbane, Australia) with parasitaemia ranging between 1–8% in sealed flasks in a gas mixture of 5% $O_2$, 5% $CO_2$ and 90% $N_2$ (BOC Gases, Brisbane, Australia) at 37°C. Medium was changed every 48 h. Cultures were routinely synchronised (typically, every 48 h), when the majority of parasites (>85%) were at ring stage using 5% D-sorbitol [48].

*In vitro* **Experiment 1.**   Synchronised ring stage culture was grown for one cycle after sorbitol treatment. The uninfected red blood cells (RBCs), which were subsequently used for making dilutions, were incubated in "complete" medium in a special gas at 37˚C for the same duration of time. Resulting ring stage culture was split into three flasks (no sorbitol treatment was carried out prior to experiment), with one used immediately for ring stage assessment, and the other two flasks were grown further to yield trophozoites and schizonts, respectively. Ring, trophozoite or schizont stage cultures and uninfected RBC were spun down at 600 x $g$ and each pellet was washed twice in "plain" media followed by another wash in 100% plasma. Infected and uninfected RBC pellets were then resuspended in 100% plasma with haematocrit adjusted to 44–48%. A total of thirteen 10-fold serial dilutions of infected RBCs were made using uninfected RBCs at 44–46% haematocrit with parasitaemia ranging from $10^6$ to $10^{-6}$ parasites/µL. Three µL of infected for each parasite stage and 10 replicates for each dilution as well as uninfected RBCs were spotted on the glass slides and dried for 1–2 h before scanning. Two independent experiments were conducted.

*In vitro* **Experiment 2.**   Synchronised ring stage *P. falciparum* D6 parasites were cultured for one cycle as described above. Uninfected RBCs were resuspended in complete media and "cultured" in the same way for 48 h preceding the experiment. Resulting culture, containing mostly ring stage parasites was centrifuged at 600 x $g$ and the media supernatant removed and reserved. Infected RBC pellet (at 5.5% parasitaemia) was washed twice in "plain" media and then split into two halves, B1 and B2. B1 pellet was subsequently washed once in 100% plasma and haematocrit measured at 40%. Pellet B2 was washed in "plain" media 10 more times before final wash in 100% plasma with haematocrit of resulting suspension measured at 46.8%. Uninfected RBCs were split into two halves U1 and U2 and washed as B1 and B2 pellets, respectively. Haematocrit of U1 and U2 pellets was measured and the values were 46.6 and 46.7% respectively. Both B1 and B2's parasitaemia were adjusted to 100,000 parasites/µL using U1 or U2 RBCs and then thirteen serial 10-fold dilutions of either of these suspensions were done using either U1 or U2 RBCs, respectively resulting in parasitaemia range of $10^5$–$10^{-7}$ parasites/µL. In addition, thirteen serial 10-fold dilutions of media supernatant were made using U1 RBCs as above. Three µL of each dilution was spotted on glass slides and scanned as described above.

*In vitro* **Experiment 3.**   To test the sensitivity of NIRS for detection of *P. falciparum* parasites cultured in RBCs originated from different donors, approximately ten mL aliquots of $O^{(+)}$ blood from each of 10 donors were additionally provided by Life blood Red Cross Service (Brisbane, Australia) in addition to the "lab" RBCs stock used for continuous culture. These 10 aliquots were washed individually: twice in "plain" media and once in "complete" media. Haematocrit was adjusted for every aliquot to 50%. *P. falciparum* D6 parasite line was synchronised as described above and grown to parasitaemia of 8%. The culture was centrifuged and the infected RBCs were used to initiate 11 new cultures (starting parasitaemia of ~0.8%), each containing the RBCs from one donor. These eleven cultures were grown for 48 h. The cultures were centrifuged and supernatants discarded. The 11 RBC culture pellets were washed twice in "plain" media and once in plasma with haematocrit measured. The uninfected RBCs from each donor were also washed in the same way as infected RBCs, with haematocrit of each suspension measured and adjusted to 44–48% range. The parasitaemia of each infected RBC pellet was adjusted to 100,000 parasites/µL and thirteen 10-fold serial dilutions were made using the respective donor's washed uninfected RBCs. As in previous experiments, three µL were spotted on the glass microscopy slides, with 10 replicates from each dilution of infected RBCs and 10 replicates of uninfected RBCs for each donor. For the two out of eleven donor samples the "blind" randomised set of blood spots was produced, where dilutions of infected RBCs and uninfected RBCs were randomly spotted on each slide and scanned.

*In vitro* **Experiment 4.**   *P. falciparum* 3D7 line was cultured and synchronized as described above. *P. falciparum* 3D7 culture at 4.3% parasitaemia of which 3.9% were rings, 0.1% were trophozoites and 0.3% were schizonts was centrifuged and the infected RBC pellet was washed 3 times as in Experiment 1. Following the final wash, the pellet was resuspended in 100% plasma with haematocrit measured at 47% with resulting parasite concentration of $2.02 \times 10^5$ parasites/μL. The uninfected RBCs were treated in same way and prepared as described above with final haematocrit adjusted to 46%. The first dilution was made to adjust parasitaemia to $10^5$ parasites/μL by diluting the pellet 2.02-fold with uninfected RBCs, followed by 18 more serial 10-fold dilutions. Ten replicates of each dilution were spotted on glass slides (3 μL per spot) starting from parasitaemia of $10^5$ parasites/μL to $10^{-13}$ parasites/μL. "Blind" sets (n = 2) of 6 slides where dilutions were randomly spotted on each slide was also made.

## *In vivo* **studies using** *P. berghei* **mouse model**

The *in vivo* studies were performed using outbred Swiss ARC female mice from the Animal Resources Centre (Canning Vale, Western Australia). The ARC mice were about 5–6 weeks old with a mean (± SD) body mass of 29.4 ± 3.5 g (n = 144). The mice were housed at about 22–24˚C with relative humidity 50 ± 15% and 12-h light/dark cycle. The animals were provided with mice maintenance pellets (Specialty Feeds, Glen Forest, Western Australia) and water *ad libitum*. The *P. berghei* ANKA line (originally from the Liverpool School of Tropical Medicine Hygiene, Liverpool, United Kingdom) was used to infect mice by intraperitoneal (IP) inoculation of donor mice blood diluted in normal saline. Each mouse was IP inoculated with $20 \times 10^6$ infected RBCs in a volume of 200 μL. Uninfected mice were inoculated with donor mice blood with saline only.

Blood samples (about 20 μL) were collected from the animal's tail vein at selected time points after IP inoculation of experimental mice with either diluent/non-parasitised uninfected RBCs or *P. berghei* infected RBCs for producing blood films for microscopy reading. The Giemsa-stained thin blood films were examined at a magnification of 1000× to determine the parasite density. Blood films were read by at least two microscopists, readings averaged and quality assurance provided by a WHO Level 1 certified malaria microscopist. At the same time that a blood sample was collected for producing blood films, two 2 μL of tail vein blood were pipetted onto a microscope glass slide to produce blood spots, allowed to dry at ambient temperature and then scanned using NIRS spectrometer.

The study design included NIRS laser scanning of mouse body areas (i.e., non-invasive assessment) and scanning of slides with blood samples collected from the tail vein of the animals (i.e., invasive assessment). To develop the NIRS model for detecting and quantifying *P. berghei* parasitaemia, the following four parts of the study were carried out:

*In vivo* **Experiment 1.**   This experiment was carried out to optimize the handling and positioning of mice for scanning selected body areas of the animal by the NIRS laser probe to produce minimal discomfort to the animals. Ten uninfected mice were allocated to the "Positioning Study". For this activity the NIRS laser probe was placed on various body areas of the mouse such as the ear, foot, groin, and tail. No adverse events such as reduction in movement, ruffled coat and loss of body mass greater than 15% of the animal's baseline value were observed during and after the "Positioning Study".

*In vivo* **Experiment 2.**   This study was carried out as a pilot study to determine whether NIRS technology could be used to differentiate uninfected from *P. berghei* infected mice, *in vivo* and to develop the NIRS infection detection model. Twenty-two mice (12 infected and 10 uninfected) were allocated to the "Pilot Study". For this part of the study, the mice body areas

and blood samples collected before (0 h) and at 24, 48 and 72 h after inoculation with either diluent (saline) or *P. berghei* parasitised RBCs were scanned with NIRS.

***In vivo* Experiment 3.**   The "Scale-Up Study" was conducted to further validate the sensitivity and precision of the NIRS model to quantitate varying levels of parasitaemia. Forty-eight mice (23 infected, 23 uninfected and two donor mice to provide non-parasitised RBCs) were allocated to the "Scale-Up Study". In the "Scale-Up Study", the mice body areas and blood samples collected before (0 h) and at 24, 48 and 72 h after inoculation with either non-parasitised uninfected RBCs or *P. berghei* parasitised RBCs were scanned with the NIRS device.

***In vivo* Experiment 4.**   The "Model Refinement Study" was designed to further refine and test the NIRS models to include low parasitaemia in mice. These mice consisted of 24 infected mice and 12 uninfected mice to refine the model and another 24 mice (12 infected and 12 uninfected) that were scanned as a blind dataset to test the NIRS models (total 60 mice). Four donor mice were also used to provide non-parasitised uninfected RBCs for inoculation of the 24 uninfected mice. In the "Model Refinement Study", the mice body areas and blood samples collected before (0 h) and at 12, 24, 35, 48 and 72 h after inoculation with either non-parasitised uninfected RBCs or *P. berghei* parasitised RBCs were scanned with NIRS.

For experiment 3 and 4 above the mice were computer randomised for the selection of animals to be infected or not infected with *P. berghei*, so the infectious status of about half of the mice were blinded to the analysts (MSL and ARL) to develop and validate the final NIRS model. Also, for all mice in Parts 3 and 4 at 72 h post inoculation they were anaesthetized and euthanized with a rising plane of carbon dioxide followed by cardiac puncture to collect about 500 μL of blood in EDTA as the anticoagulant to produce at least 20 replicate blood spots of 2 μL each, allowed to dry and scanned to obtain NIRS replication data.

## Ethics

The study was conducted in compliance with the animal ethics guidelines to promote the well-being of animals used for scientific purposes, National Health and Medical Research Council 2008 (Australian code for the care and use of animals for scientific purposes, NHMRC 8th Edition 2013). The Defence Animal Ethics Committee reviewed and approved the animal experimental protocol DAEC 09–17 for this study.

### Acquisition of spectral data using NIRS

The Labspec 4i NIRS spectrometer with an external bifurcated fiber optic probe with 6 illumination fibres whose wavelengths range from 350–2500 nm (Malvern ASD Panalytical, Malvern, UK) was used throughout the study. To scan the blood spots on glass slides prepared from *in vitro* cultures and blood samples obtained from mice, the probe was placed 2 cm above the sample.

For non-invasive scanning infected and uninfected mice were randomly scanned using the Labspec 4i NIRS spectrometer. The four different body areas (ear, foot, groin and tail) of each experimental mouse were scanned by placing the probe so that it was in direct contact with the animal's body area. Two spectral signatures were collected from each body area resulting in a total scan time of approximately 35 seconds per mouse. Each spectral signature collected was an average of 15 spectral scans. Spectral signatures were collected from dry blood spots obtained from mice using the same spectrometer. The scanning procedure used for blood spots and mice and the associated NIRS raw spectra is shown in Fig 1.

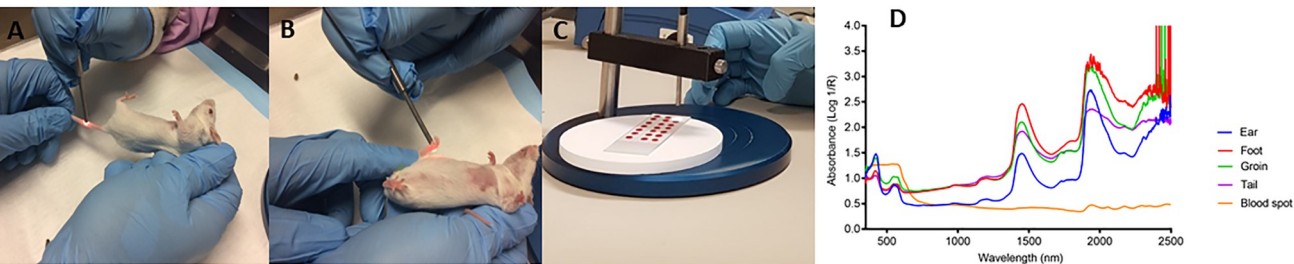

**Fig 1. Scanning of mice and blood spots using NIRS.** Panel A and B illustrates how mice were held and non-invasively scanned. Panel C illustrates scanning of dry blood spots on the slides. Panel D shows the resultant raw spectral signatures from various body parts of a mouse and a spectral signature from blood spots.

## Data analysis

Spectra data were converted to csv format and analyzed using JMP Pro version 16 (SAS Institute Inc., Cary, NC, 1989–2021). Predictive analytics were developed using artificial neural networks (ANN). These networks were fully connected, consisting of an input layer (400 nodes), 1 hidden layer (3 nodes) and an output layer (1 node). Activation between nodes was performed using the TanH function. All input nodes specified a different frequency between 500 and 2350 nm with a step size of 1 nm. The output layer was trained as either a binary outcome (e.g., infected *vs* uninfected; rings *vs* schizonts) or continuous outcome (e.g.,parasitaemia level). Different models were trained for each test resulting in different weights, although the network structure remained unchanged.

Spectral signatures collected from dried spots derived from *P. falciparum in vitro* cultures and spectral signatures collected non-invasively and invasively using *P. berghei* mouse model were analyzed using similar functions of predictive neural networks but separately.

For *in vitro* cultures, ANN models were developed using rings, trophozoites and schizont stages grown in experiment 1. The resultant model was then applied to predict samples from experiment 1 that were excluded from the model and samples that were "blind" to the analyst (Validation set). This model was further validated on independent samples grown at three different time points referred to as experiment 2 (test set 1), experiment 3 (test set 2) and experiment 4 (test set 3) (see cultivation of *in vitro* cultures above). To refine the model, a subset of samples from each experiment were subsequently added into the original model and the resultant model was applied to predict the remaining samples. A second model was developed to differentiate the three malaria stages and the resultant model was used to predict samples excluded from the model and samples unknown to the analyst.

For the *in vivo* studies, the ANN model was developed to predict infection status and parasitaemia level non-invasively in 46 mice randomly selected from the three experiments described above. Mice were randomly selected from each of the four experiments conducted to increase variability hence increasing the robustness of the model. The parasitaemia level of samples used in the model varied from 0.06–26.7%. The mice that were used to train the model were split into training set, validation set and test set. The second model predicted infection and parasitaemia of mice using blood samples drawn from mice (n = 63). Similarly, spectral data was split into the training set, validation set and test set. All models were further validated using independent test sets (i.e., samples blinded to the model and samples blinded to the analyst).

A flow chart of the number of samples screened by NIRS that includes both the training, validation and test sets for the various *in vitro* and *in vivo* experiments are sumarised in supplementary S1 Fig.

The spectral region analyzed included the NIRS optical window with bands between 500–1100 nm where maximum penetration of light in biological samples occurs to include the haemoglobin absorption regions and exclude water related bands. To ensure the robustness of models, K-fold cross validation (K = 5) was employed. Regression analysis was used to assess the correlation between parasitaemia estimated by blood film microscopy and NIRS predicted parasitaemia.

## Results

### Detection of asexual blood stages of *P. falciparum* parasites *in vitro* using NIRS

The first experiment was designed to develop a machine learning model that could differentiate RBCs infected with *P. falciparum* D6 line from uninfected RBCs regardless of the parasite stage and parasitaemia. The infected RBCs derived from *P. falciparum* D6 culture were washed and resuspended in human plasma to 46% haematocrit to mimic "real" blood composition. Thirteen serial 10-fold dilutions were made using uninfected RBCs (also at 46% haematocrit in human plasma) used to culture parasites and originated from a single human donor. This training model was 100% accurate for differentiating infected from uninfected samples excluded from the model (n = 352). The validation set included the samples (n = 820) collected from the same experiment as the samples used to train the model, which were "blind" to the model and the analyst. The training model was 99.6% accurate for differentiating infected (n = 760) from uninfected RBCs (n = 60) with a sensitivity of 99.6% and specificity of 100% at parasite densities as low as $10^{-7}$ parasites/μL.

To further validate the accuracy of the model developed above, we conducted the second experiment (test set 1) to determine whether the products detected by NIRS are present in the supernatant or associated with infected RBCs. We compared the sensitivity of NIRS detection of *P. falciparum* D6 malaria parasites derived from *in vitro* culture after washing the infected RBC pellet 3 times (as in the previous experiment) against that washed 13 times. Thirteen ten-fold serial dilutions of the infected RBC ranging from $10^5$–$10^{-7}$ parasites/μL were made using uninfected RBCs (46% haematocrit in human plasma) from the same donor. Regardless of the parasitaemia level, the training model was 86% sensitive (n = 479), 89% specific (n = 90) and 89% (n = 569) accurate for detecting *P. falciparum* when pellets were washed three times. This sensitivity increased to 93% (n = 480) but specificity dropped to 14% (n = 90), respectively with additional 10 washes prior to diluting the parasites. The sensitivity of the model was not dependent on the parasitaemia of samples. We also tested the culture media supernatant that was removed after the first centrifugation of infected RBCs step before subsequent washing. The model developed was 62% sensitive (n = 520) in detecting the presence of malaria parasites byproducts in the supernatant diluted up to $10^{-7}$ folds.

In the third experiment (test set 2), we tested the ability of the refined model to detect malaria parasites in RBCs from 11 different human donors. In this experiment, *P. falciparum* D6 parasites were cultured in the RBCs from a single donor to parasitaemia of 8%. These stock RBCs were used to inoculate 11 new cultures containing the original RBCs, as well as RBCs from 10 different donors. The stock RBCs from the initial culture were diluted (1:10 ratio) with starting parasitaemia of 0.8%. The cultures were allowed to grow for 48 h to achieve parasitaemia of 7.8 ± 1.0% (average ± standard deviation). RBC pellets from 11 cultures were washed 3 times and subsequently serially diluted as above with the uninfected RBCs used in

**Table 1. Sensitivity, specificity ad accuracy of NIRS for predicting malaria parasites *in vitro* at various treatments regardless of the parasitaemia.**

| Treatment | Validation | | | Test set 1 | | | Test set 2 | | | Test set 3 | | |
|---|---|---|---|---|---|---|---|---|---|---|---|---|
| | Sen [N] | Spec [N] | Acc [N] | Sens [N] | Spec [N] | Acc [N] | Sen [N] | Spec [N] | Acc [N] | Sens [N] | Spec [N] | Acc [N] |
| 3 washes | 99.6 [760] | 100 [60] | 99.6 [820] | 86 [479] | 89 [90] | 89 [569] | 96 [911] | 93 [90] | 96 [1001] | 96 [130] | 93 [40] | 96 [170] |
| 13 washes | - | - | - | 93 [480] | 14 [90] | 79 [580] | | | | | | |
| Supernatant | - | - | - | 62 [520] | | | | | | | | |

Sen: Sensitivity; Spec: Specificity; Acc: Accuracy; N: Number of Samples

*Validation Set*: Samples were collected at the same time as the samples used to develop the model

*Test set 1.* *P. falciparum* D6 line spectra of 3 washes *vs* 12 washes collected in 2021

*Test set 2.* *P. falciparum* D6 spectra of samples collected from multiple donors in 2021

*Test set 3.* *P. falciparum* 3D7 spectra collected in 2022

the respective cultures to achieve concentrations ranging from $10^5$–$10^{-7}$ parasites/μL. Irrespective of the parasitaemia and the RBCs donor, the refined model, which included a small proportion of samples (n = 3 donors) from this experiment was 96% sensitive (n = 911), 93% specific (n = 90) and 96% accurate (n = 1001) for differentiating infected from uninfected samples. The model was also tested using "blind" set, containing randomly spotted dilutions of parasites cultures, in which blood from two donors was used.

In the fourth experiment (test set 3) we used 3D7 *in vitro P. falciparum* cultures to test the accuracy of the model. The refined model was also 96% sensitive (n = 130), 93% specific (n = 40) and 96% accurate (n = 170) for differentiating infected from uninfected samples in all 19 dilutions, ranging from $10^5$ to $10^{-13}$ parasites/μL. In all three experiments, we have not shown the effect of various parasite densities on the prediction of NIR because of the high sensitivity and specificity observed across all densities. A summary of results of independent samples used to validate the training model for all four experiments is presented in Table 1.

## Differentiation and quantification of asexual stages *P. falciparum in vitro*

A separate model was trained using artificial neural network technique to differentiate the three asexual blood stages of *P. falciparum*: rings (n = 416), trophozoites (n = 416) and schizonts (n = 384). The model differentiated the three parasite stages from each other with 100% accuracy. When using the validation sets containing rings (n = 104), trophozoites (n = 104) and schizonts (n = 96), the model was 100% accurate. For the independent test sets (i.e., the data set that was blind to the model and the analyst), NIRS was 98% (n = 820) accurate for differentiating the three blood asexual stages. It differentiated rings from trophozoites and schizonts with 99.6% accuracy (n = 281), trophozoites from rings and schizonts with 99% accuracy (n = 280) and schizonts from rings and trophozoites with 96.5% accuracy (n = 259). A summary of how NIRS differentiated the three blood stages of *P. falciparum* is shown in Fig 2.

Our last experiment involved quantifying parasitaemia of the three parasite stages across a wide range of parasite densities ($10^5$ to $10^{-7}$). ANN produced correlation coefficient values ($R^2$) of 0.71, 0.81 and 0.85 for schizonts, rings and trophozoites, respectively. However, NIR failed to quantify parasitaemia in dilutions of samples from the 11 donors which contained different blood stages of *P. falciparum* indicating potential confounding factors from multiple donors that should be investigated in future studies. S2 Fig shows the correlation plots between actual parasitaemia measured by microscopy plotted against NIR predicted parasitaemia of schizonts, rings and trophozoites. The results shown include samples that were excluded from the training/validation set and were blind to the analyst (i.e., the test set).

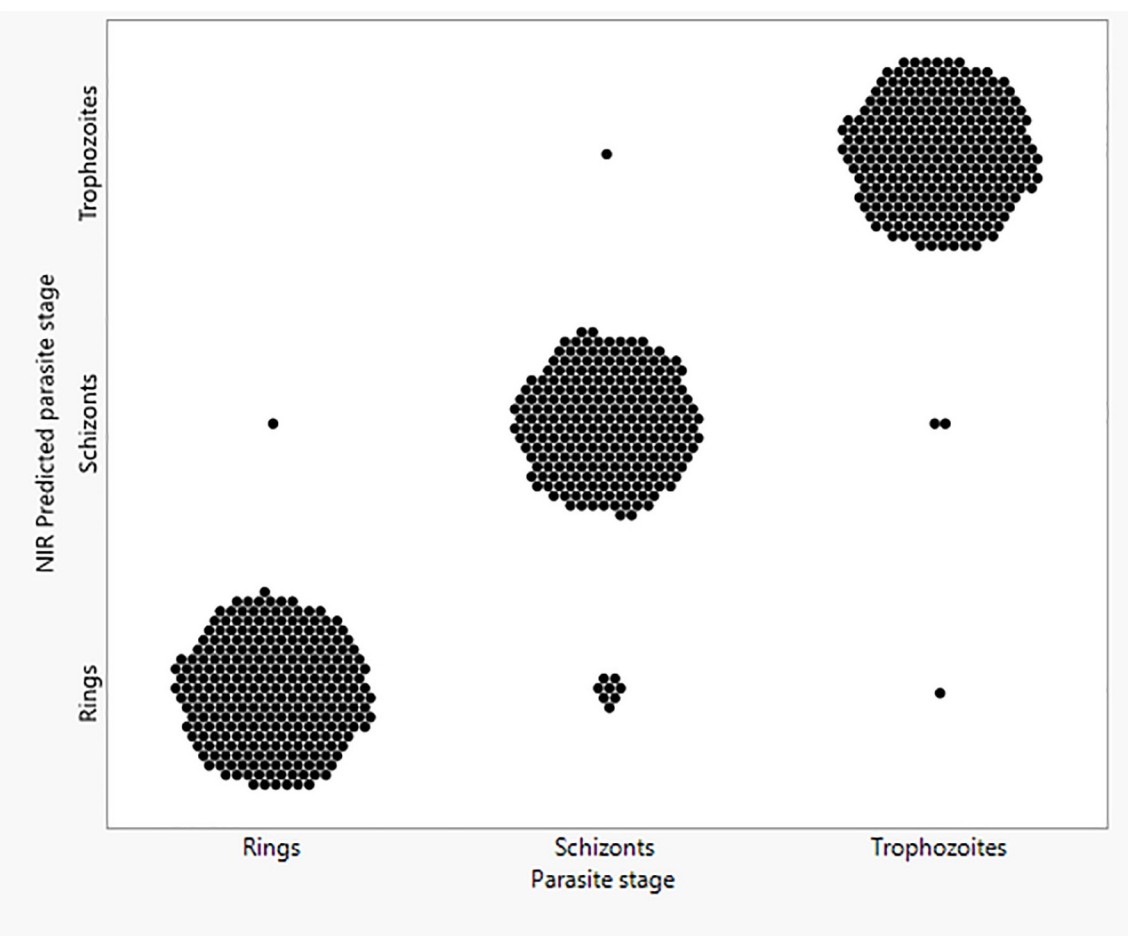

**Fig 2. NIRS differentiation of *P. falciparum* rings, schizonts and trophozoites.** Each dot represents one replicate for the indicated stage.

## Detection of *P. berghei* infection in mice using NIRS

NIRS was evaluated for *in vivo* detection and quantification of *P. berghei* parasites in mice, non-invasively, by scanning the ear, the groin, the foot and the tail of infected and uninfected mice or invasively, by scanning blood spots collected from mice on microscopy slides. The results were validated by microscopy and Giemsa stained thick and thin blood smears by a WHO certified technician. Overall, NIRS was more sensitive when used non-invasively than when it was used to scan blood spots from mouse tail snips. However, the sensitivity of NIRS for both non-invasive and invasive prediction of malaria parasites was highly dependent on parasitaemia of the mouse at the time of scanning. In both cases, the prediction accuracy increased with rising parasitaemia levels. For example, infections in mice with parasitaemia <0.5% were least accurately predicted with sensitivity ranging from 42–65% whereas infections in mice with parasitaemia >3% were the most accurately predicted with sensitivities ranging from 80–95% depending on type of scanning and the body part scanned.

Spectral signatures acquired from the foot produced the highest prediction accuracy whereas spectra from the tail produced the least accurate predictions. For example, at parasitaemia levels of ≤0.5%, 0.6–2% and ≥3%, NIRS was 59%, 90% and 91% sensitive, respectively,

**Table 2. The sensitivity of NIRS for predicting *Plasmodium* infection either in blood spots or non-invasively by scanning ears, feet, groins and tails of mice used to test the accuracy of the artificial neural network model developed to predict infection.** The table also shows the effect of parasitaemia on NIRS prediction accuracy using non–invasive and blood spots.

| Parasitaemia | Scanning blood spots | | Non-invasive scanning of various body areas | | | | | | | |
|---|---|---|---|---|---|---|---|---|---|---|
| | Sensitivity (%) [N*] | Specificity (%) [N†] | Sensitivity (%) [N*] | | | | Specificity† (%) [N] | | | |
| | | | Ear | Foot | Groin | Tail | Ear | Foot | Groin | Tail |
| ≥3% | 95 [38] | 84 [213] | 94 [66] | 91 [66] | 92 [66] | 80 [66] | 86 [342] | 87 [342] | 79 [342] | 82 [342] |
| 0.6–2% | 63 [36] | 84 [213] | 71 [48] | 90 [48] | 81 [48] | 62 [48] | 86 [342] | 87 [342] | 79 [342] | 82 [342] |
| ≤0.5% | 46 [38] | 84 [213] | 42 [72] | 59 [72] | 65 [72] | 53 [72] | 86 [342] | 87 [342] | 79 [342] | 82 [342] |
| Overall** | 86 [112] | 84 [213] | 70 [179] | 80 [179] | 82 [179] | 66 [179] | 86 [342] | 87 [342] | 79 [342] | 82 [342] |

*Number of mice, for which parasitaemia was at the level indicated at the time of scanning

† Number of mice that were not infected at the time of scanning

** Provides overall results for sensitivity and specificity regardless of the parasitaemia level

when spectral signatures from the foot were analysed (Table 2). Comparatively, when blood spots were scanned at similar parasitaemia, sensitivity values of 46%, 63% and 95% were achieved, respectively. NIRS was 94% sensitive, when parasitaemia levels were ≥3% with spectra collected from the ears. This sensitivity was comparable to a sensitivity of 95% achieved when NIRS was used to scan blood spots at the same parasitaemia.

Regardless of the parasitaemia level, NIRS non-invasively predicted the presence of parasites in mice with an overall sensitivity of 66%, 70%, 80% and 82% and a specificity of 82%, 86%, 87% and 79% when the tail, ear, foot and groin, respectively, were scanned. Comparatively, when blood spots were scanned, an overall sensitivity and specificity of 84% and 86%, respectively, was achieved. The effect of parasitaemia on NIR sensitivity is shown in Fig 4 and the summary of NIRS prediction at various parasitaemia levels is shown in Table 2.

A positive and significant correlation was observed between NIRS predicted parasitaemia and parasitaemia estimated by microscopy for both non-invasive scanning ($R^2$ = 0.55

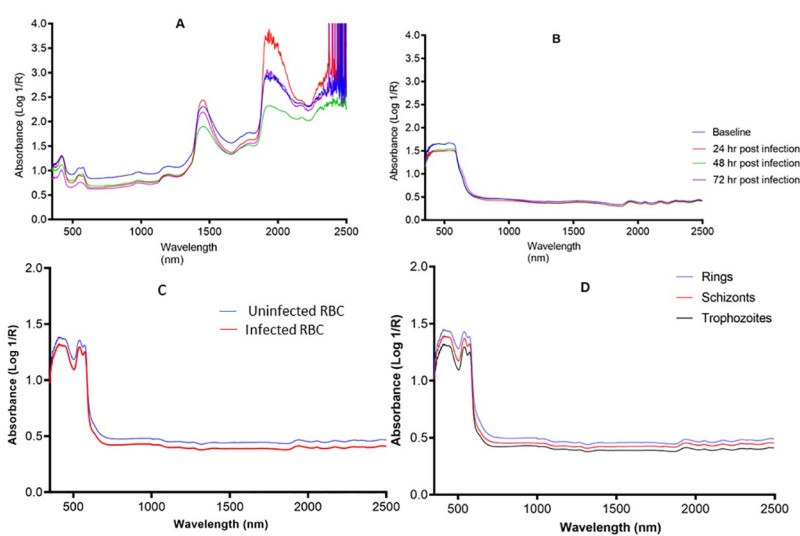

**Fig 4. Raw NIRS spectra data from *in vivo* mice and *in vitro* culture studies.** Change in average raw spectra as percentage parasitemia increases from 0 at baseline to 0.41, 3.74 and 20.33 post-inoculation for the spectra collected from mice non-invasively (A), red blood cells (B) *in vitro* cultures (Panel C) and malaria parasite stages (D).

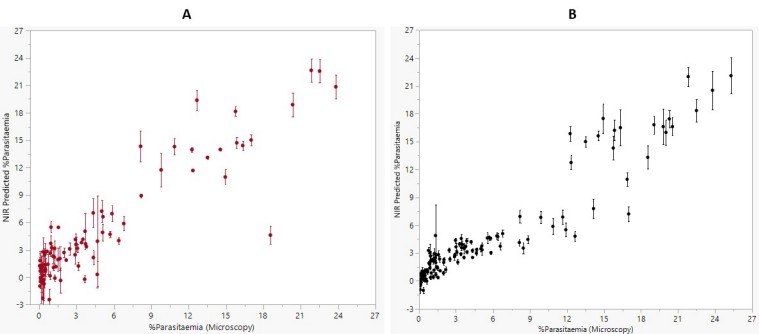

**Fig 3. NIRS prediction of *P. berghei* infection in mice.** NIRS prediction of parasitemia levels relative to microscopy for mice that were scanned invasively (A) and non-invasively (B). Dots represent mean and error bars indicate standard error of the mean.

$P<0.001$) and for blood spots ($R^2 = 0.76$ $P<0.001$) NIRS was more accurate for predicting parasitaemia at higher levels than at lower parasite densities (Fig 3). The effect of parasitaemia on the sensitivity of NIRS for both non-invasive and invasive scanning is shown in S3 Fig.

## Raw spectra

A difference was observed in the absorption values of the raw spectra collected either non-invasively from mice or from blood spots from mice and *in vitro* cultures scanned at various time points where parasitaemia was observed to increase. Generally, the absorbance spectra of infected mice or infected blood spots was lower than the absorbance of uninfected samples for both non-invasive (Fig 4A) and invasive scans (Fig 4B and 4C). The absorption spectra of infected samples also tended to cluster together away from absorption spectra of uninfected samples for mice that were scanned over a period of 72 hours (Fig 4A). Spectra of rings, tro-phozoites and schizoints could also be differentiated (Fig 4D).

## Second derivative graphs

Peaks of importance were observed from second derivative graphs with 10 smoothening points for spectra acquired non-invasively, blood collected from tails of mice and blood spots from *in vitro* cultures in the visible region and the near-infrared region. Fig 5 shows second derivative graphs for *P. berghei* infected and uninfected mice scanned non-invasively (Fig 5a and 5d) and in mice blood samples (Fig 5b and 5e) and second derivative spectra of *P. falciparum* infected and uninfected RBCs in *in vitro* cultures (Fig 5c and 5f) for visible (a, b, c) and the near-infra-red region (d, e, f). Within the visible region, an absorption band corresponding to hemozoin was observed at 653 nm for *P. berghei* parasites in spectra acquired non-invasively in mice (Fig 5a and 5d) and at 654nm for blood spots collected from mice (Fig 5b). For *in vitro* blood spots, a hemozoin band was observed around 670 nm. Previous studies identified hemozoin absorp-tion peaks at 643 nm, [40], 650 nm [41] and 655 nm [42]. Within the near-infrared region (Fig 5d–5f), absorption bands around 1737, 1984 and 2216 nm observed *in vitro* cultures and blood collected from infected mice (Fig 5b) are also related to hemozoin protein and these peaks have also recently been observed in two other studies [38, 41].

An absorption band for haemoglobin was observed within the visible region at 690 nm in all samples from all three experiments (Fig 5a). Other haemoglobin related bands were observed at 1696, 2168, 2290 and 2054 nm as seen in spectra from *in vitro* cultures and these bands have also been previously reported [38, 43].

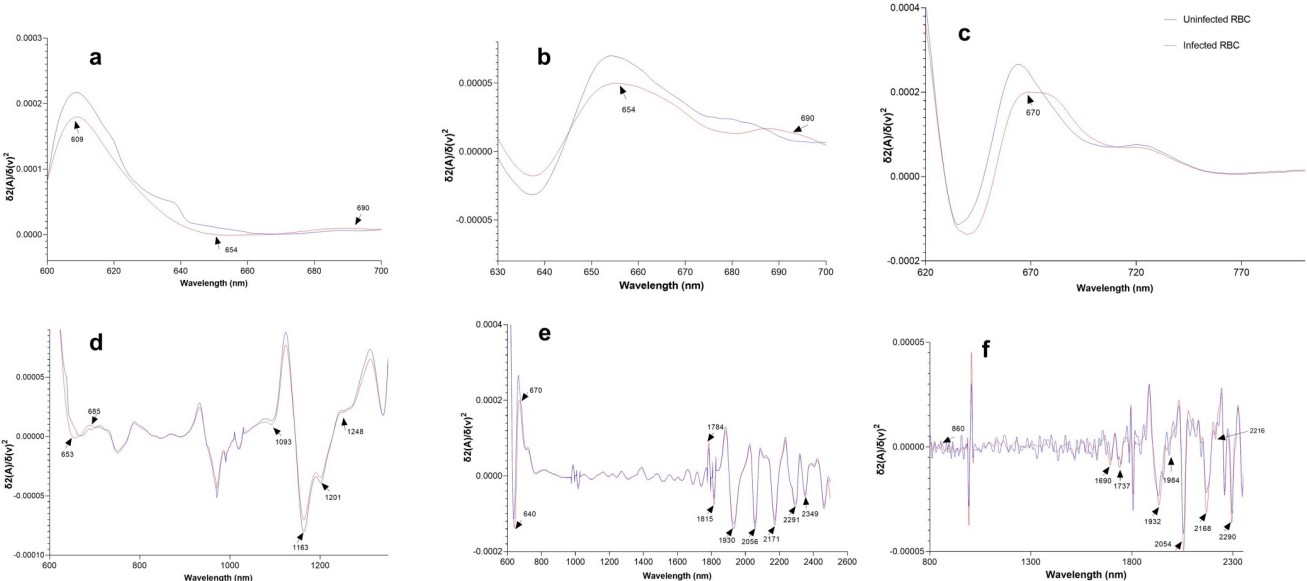

**Fig 5. Second derivative spectra of the visible region (a-c) and the visible-NIR region (d-f) showing absorption peaks for hemozoin and hemoglobin protein for spectra acquired non-invasively (a and d), in blood spots collected from mice (b and e) and *in vitro* blood spots (c and f).**

## Discussion

In this study, we demonstrated NIR spectra coupled with machine learning allowed the detection, quantification and differentiation of blood asexual stages of *P. falciparum* parasites. These *in vitro* findings provided the impetus to evaluate the potential of NIRS to non-invasively detect rodent malaria as a proof-of-concept for malaria diagnosis in human studies. We showed using the ANN model, the detection and quantification of *P. falciparum* in *in vitro* cultures at various parasite densities, *P. berghei* infections in mice either non-invasively (i.e., ear, foot, groin and tail) or invasively (i.e., tail vein blood sample). To our knowledge, this study represents the first investigation of a comparative evaluation of NIRS for *in vivo* and *in vitro* application as a potential diagnostic and surveillance tool for malaria.

NIR predicted the presence or absence of *P. falciparum* malaria parasites and malaria parasites byproducts with high sensitivities when blood from a single donor and multiple donors was used to culture and make dilutions of the infected RBCs. The ability of NIRS to detect malaria at this very low parasitaemia of $10^{-7}$ parasites/μL is an indicator that it could potentially be a useful diagnostic tool for asymptomatic and submicroscopic malaria carriers. This parasitaemia level was between 7 and 8 orders of magnitude more sensitive than the limit of detection for microscopy [11] and RDTs [44, 45]. This suggests that the spectral signatures used to detect and quantify malaria parasites are likely dependent on the malaria parasite byproducts (e.g., biocrystal hemozoin) as opposed to the presence of parasites in RBCs as it is highly likely that some of the blood samples correctly predicted as infected did not contain any malaria parasites. The latter is also consistent with detection of the parasite byproducts after washing the parasite pellet 13 times as well as in the lowest dilutions of culture media. It is plausible that these byproducts are excreted from the parasites and abundantly present in culture media or mouse blood. It also follows that the abundance of these products is much greater than that of DNA or even mRNA molecules used for parasite detection by the most sensitive currently available RT-qPCR methods, with a limit of detection of $10^{-2}$–$10^3$ parasite/

µL. This finding also indicates that the spectral signatures used to detect and quantify malaria parasites are dependent on the secretion of these byproducts and the malaria parasite biocrystal, hemozoin as opposed to the presence of the parasite itself in RBCs. Field studies in a low transmission setting with asymptomatic malaria patients could provide an insight into the diagnostic capacity of NIRS in real world settings, and its utility in malaria elimination programs.

Previous studies of the parasite proteome revealed a large number of unique proteins and other metabolites present at every stage of the parasite life-cycle [46–48]. When NIRS light interacted with each of the three parasite blood asexual stages, unique spectral signatures with varying absorbance values were observed (Fig 5). The ability of NIRS to differentiate the three stages could be due to the fact that each stage of the malaria parasite is morphologically and biochemically distinguishable. Rings, trophozoites and schizont stages were differentiated from each other with 98% accuracy. Our findings show that NIRS accurately identified all samples at the rings stage (n = 281) except one and differentiated rings from schizonts with 98% accuracy (n = 540) and rings from trophozoites with 99% accuracy (n = 540). In particular, the ability to detect ring stages is key to early diagnosis of *P. falciparum* malaria. This is because ring stages are predominant in the peripheral blood of people infected with *P. falciparum*, whereas trophozoites and schizonts sequester and adhere to microvasculature of the brain and other organs.

The sensitivity of NIRS for detecting *P. berghei* non-invasively and invasively from mice was highly influenced by the parasitaemia of the mouse at the time of scanning. Surprisingly, NIRS was generally more sensitive when used non-invasively than when scanning blood samples collected from the tail of mice. This could be because there is minimal circulation of blood to the tail of the mice hence less parasites/parasite byproducts are present when blood is drawn from the tail. Moreover, compared to other body parts scanned, the tail produced the least accurate prediction. The higher predictive accuracy seen in cultured *P. falciparum* at low parasitaemia levels could be due to the presence of malaria parasite byproducts produced during culturing. It is recommended that future studies assess the capacity of the technique to detect infection using non-invasive samples such as urine or saliva.

Non-invasive diagnosis could significantly reduce the current cost and time required for malaria diagnosis allowing routine diagnosis to be performed rapidly on a larger scale or even at a household level to guide the current malaria elimination strategies. Moreover, NIRS does not require running costs nor sample processing procedures after the initial purchase of the NIRS instrument. For this study, it took on average 5 seconds to acquire a diagnostic spectrum from blood spots or non-invasively from a mouse. Following development of diagnostic algorithms, diagnosis of thousands of samples is instantaneous. The use of NIRS as a diagnostic tool can be fully automated either via a cloud-based system or by loading prediction models on spectrometers allowing unskilled personnel to perform real time diagnosis even in remote areas or anywhere around the world. Comparatively, current malaria diagnostic techniques require a finger prick capillary blood sample and at least a two-step sample processing procedure that involves blood film staining followed by interpretation of the results by competent microscopists. This limits the number of samples that can be processed and read by a technician in a day to about 60 blood films. Comparatively, NIRS can potentially analyse hundreds of samples per day per technician.

We have shown that NIRS spectra collected from multiple body areas of mice produce varying degrees of infection detection. For example, the ear and the foot produced the cleanest spectra and the most accurate prediction of the presence or absence of *P. berghei* infection. Comparatively, spectra from the tail were the noisiest and produced the least accurate results. Unsurprisingly, the sensitivity of NIRS in detecting *P. berghei* when the animal's tails were

scanned was similar in blood samples collected from the tail vein. This could imply less parasites are sequestered in the tail relative to other body areas of the mouse.

Compared to uninfected samples, there was a general decrease in absorption spectra of infected *in vitro* cultures of *P. falciparum* and infected mice that were scanned non-invasively and invasively (Fig 5). This may suggest a reduction in haemoglobin concentration among infected RBCs compared with uninfected RBCs. Similar results were observed by Adegoke and colleagues using a *P. falciparum* line, *in vitro* [38, 41]. The region analysed (500–1100 nm) is commonly referred to as the optical window or the NIRS shortwave window. Within this window, there is maximum penetration of light in biological tissues. The region also incorporates haemoglobin absorption bands and bands of C-H, O-H, N-H from the third overtone region.

The development of a point-of-care rapid non-invasive infrared based technique is in line with the current urgent need to accelerate malaria elimination, particularly in identifying people with asymptomatic malaria for targeted drug therapy. As a proof-of-concept, we have demonstrated the potential of NIRS to non-invasively detect rodent malaria *in vivo* and in blood spots, its ability to differentiate malaria parasite stages and quantify the stages which now requires a full-scale field evaluation and validation in human subjects infected with malaria particularly in areas with asymptomatic or submicroscopic malaria population. Assessment of the technique's capacity to detect other malaria parasites including hemolysed samples based on this protocol should be considered in future work.

## Supporting information

**S1 Fig. All data used in the *in vivo* and *in vitro* studies and how it was distributed between the training, validation and test set for machine learning.**
(TIF)

**S2 Fig. NIRS quantification graphs for trophozoites, rings and schizonts.**
(TIF)

**S3 Fig. Effect of parasite densities on predicting infection in mice.** The effect of parasitemia for non-invasive (Panel A) and invasive (Panel B) prediction of *P. berghei* infection, where mice with high parasitemia levels are more likely to be predicted as infected compared to mice with lower parasitemia values.
(TIF)

## Acknowledgments

We acknowledge the technical excellence of Ms. Kerryn Rowcliffe for *in vitro* culturing and we thank Mr Stephen McLeod-Robertson, Mr Anthony Kent and Ms. Karin Van Breda for carrying out the *in vivo* mice studies, as well as Dr Ivor Harris for quality assurance of microscopy readings. We also acknowledge the technical assistance provided by Dr Jill Fernandes and Ms. Tharanga Kariyawasam and statistical advise by Dr Paul Muhindira. We are grateful to the Australian Red Cross LifeBlood Service for the provision of human blood and plasma for *in vitro* cultivation of the *P. falciparum* line. The opinions expressed are those of the authors and do not necessarily reflect those of the Australian Defence Organisation and United States Department of Agriculture or any extant policy.

## Author Contributions

**Conceptualization:** Maggy T. Sikulu-Lord, Michael D. Edstein, Floyd E. Dowell, Marina Chavchich.

**Data curation:** Maggy T. Sikulu-Lord, Brendon Goh, Anton R. Lord.

**Formal analysis:** Maggy T. Sikulu-Lord, Brendon Goh, Anton R. Lord.

**Funding acquisition:** Maggy T. Sikulu-Lord, Michael D. Edstein, Anton R. Lord, Geoffrey W. Birrell, Marina Chavchich.

**Investigation:** Maggy T. Sikulu-Lord, Michael D. Edstein, Brendon Goh, Jye A. Travis, Marina Chavchich.

**Supervision:** Maggy T. Sikulu-Lord.

**Writing – original draft:** Maggy T. Sikulu-Lord.

**Writing – review & editing:** Maggy T. Sikulu-Lord, Michael D. Edstein, Brendon Goh, Anton R. Lord, Jye A. Travis, Floyd E. Dowell, Geoffrey W. Birrell, Marina Chavchich.

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
