## [Decision Letter · Decision Letter 0]

29 Aug 2023

PONE-D-23-19957­­­Rapid and non-invasive detection of malaria parasites using near-infrared spectroscopy and machine learningPLOS ONE

Dear Dr. Sikulu-Lord,

Thank you for submitting your manuscript to PLOS ONE. After careful consideration, we feel that it has merit but does not fully meet PLOS ONE’s publication criteria as it currently stands. Therefore, we invite you to submit a revised version of the manuscript that addresses the points raised during the review process.

We look forward to receiving your revised manuscript.

Kind regards,

Yash Gupta, Ph.D.

Academic Editor

PLOS ONE

“M.S-L., M.D.E. A.R.L., G.W.B., F.E.D., M.C; APP 1159384; National Health and Medical Research Council, Australia; https://www.nhmrc.gov.au/funding  

M.S-L.; AQIRF019-2018; Advance Queensland Industry Research Fellowship; https://advance.qld.gov.au/grants

Australian Defence Organisation”

6. We note that Figure 1 in your submission contain copyrighted images. All PLOS content is published under the Creative Commons Attribution License (CC BY 4.0), which means that the manuscript, images, and Supporting Information files will be freely available online, and any third party is permitted to access, download, copy, distribute, and use these materials in any way, even commercially, with proper attribution. For more information, see our copyright guidelines: http://journals.plos.org/plosone/s/licenses-and-copyright.

Additional Editor Comments:

Authors need to thoroughly revise the manuscript with updated references as suggested by the reviewers. Species specificity and methodology gaps need to be properly addressed in the revised manuscript.

Reviewers' comments:

Reviewer's Responses to Questions

**Comments to the Author**

1. Is the manuscript technically sound, and do the data support the conclusions?

Reviewer #1: Yes

Reviewer #2: Yes

2. Has the statistical analysis been performed appropriately and rigorously? 

Reviewer #1: Yes

Reviewer #2: N/A

3. Have the authors made all data underlying the findings in their manuscript fully available?

Reviewer #1: Yes

Reviewer #2: Yes

4. Is the manuscript presented in an intelligible fashion and written in standard English?

Reviewer #1: Yes

Reviewer #2: Yes

5. Review Comments to the Author

Reviewer #1: This research work has its own importance towards malaria diagnosis. The scientific group tried here to show the importance and utilisation of near-infrared-spectroscopy (NIRS) for diagnosis of malaria parasite (Plasmodium falciparum and P. berghei). The findings are interesting. However, the design and execution of experiments doesn’t seem adequate to establish the efficient use of NIRS for detection of malaria pathogens. The research work needs major revision before its publication. Below are few concerns observed;

Minor Revision

Citations seems inappropriate. Many of the below citations can be omitted/replaced by predecessor relevant citations.

1. Line 87 bear citation [7-9], which seems additional as this can be cited under earlier citation [3-6]. In addition reference-7 is not related to the author’s thoughts.

2. Similarly in line 93-95 citation [12] can be cited as [11] and in [13-15] citation [13,14] are not required and reference-15 is adequate for the statement.

3. Similarly, citation [16-18] can be replaced by reference-11 in line 100-101.

4. Line 107-108 can be rephrased as ultrasensitive quantitative PCR under citation [23] only and no need of citation [24-25]

5. Line 110 has citation [32] not related to LAMP for malaria diagnosis, so it can be omitted.

6. Citation [34] in line 115 is improper, however this citation is ok at other statements.

This research work has its own importance towards malaria diagnosis. The scientific group tried here to show the importance and utilisation of near-infrared-spectroscopy (NIRS) for diagnosis of malaria parasite (Plasmodium falciparum and P. berghei). The findings are interesting. However, the design and execution of experiments doesn’t seem adequate to establish the efficient use of NIRS for detection of malaria pathogens. The research work needs major revision before its publication. Below are few concerns observed;

Minor Revision

Citations seems inappropriate. Many of the below citations can be omitted/replaced by predecessor relevant citations.

1. Line 87 bear citation [7-9], which seems additional as this can be cited under earlier citation [3-6]. In addition reference-7 is not related to the author’s thoughts.

2. Similarly in line 93-95 citation [12] can be cited as [11] and in [13-15] citation [13,14] are not required and reference-15 is adequate for the statement.

3. Similarly, citation [16-18] can be replaced by reference-11 in line 100-101.

4. Line 107-108 can be rephrased as ultrasensitive quantitative PCR under citation [23] only and no need of citation [24-25]

5. Line 110 has citation [32] not related to LAMP for malaria diagnosis, so it can be omitted.

6. Citation [34] in line 115 is improper, however this citation is ok at other statements.

Reviewer #2: Manuscript is well written and data are well presented. Authors have addressed an interesting and essential aspect which needs to be addressed. The need for a non-invasive methodology is necessary in malaria endemic regions for quick screening of infected and uninfected subjects. The study has pointed out that NIS can screen the parasite accurately in limited time. Using different experimental setup, the authors have also shown the efficiency of this technique. I find the article interesting and a step forward in rapid and robust screening parasite on large scale in short frame of time.

Comments:

1) If possible, the authors may include hemolyzed samples for efficiency of this methodology for detection.

2) Further, authors have shown that the higher parasitemia is detected more accurately in vivo system this suggests that the lower limit of detection in vivo and in vitro is different. Authors may discuss the factors which may be responsible for this difference.

3) It would have been more interesting to see the detection of malaria parasite in the infected individuals without any processing. The parasitemia in individual usually not high except under chronic condition or in highly endemic regions.

4) Is this methodology specifically related to Plasmodium falciparum or to other malarial parasite?

5) Authors assert that “specific components in biological samples” be responsible for the spectra change if true would spectra of urine from infected mice samples may also show change in spectra?

6) Throughout the study O+ blood was used is the data. Does the detection accuracy similar in all other blood groups?

Authors may discuss above points throughout the manuscript. Overall, I recommend this article for publication.

6. PLOS authors have the option to publish the peer review history of their article (what does this mean?). If published, this will include your full peer review and any attached files.

Reviewer #1: **Yes: **Raja Babu Singh Kushwah

Reviewer #2: No

---

## [Author Response · Author response to Decision Letter 0]

6 Dec 2023

Responses to specific reviewer's comments can be found in the attached 'Response to reviewers' document.

---

## [Decision Letter · Decision Letter 1]

28 Dec 2023

­­­Rapid and non-invasive detection of malaria parasites using near-infrared spectroscopy and machine learning

PONE-D-23-19957R1

Dear Dr. Sikulu-Lord,

We’re pleased to inform you that your manuscript has been judged scientifically suitable for publication and will be formally accepted for publication once it meets all outstanding technical requirements.

Kind regards,

Yash Gupta, Ph.D.

Academic Editor

PLOS ONE

Additional Editor Comments (optional):

Reviewers' comments:

Reviewer's Responses to Questions

**Comments to the Author**

1. If the authors have adequately addressed your comments raised in a previous round of review and you feel that this manuscript is now acceptable for publication, you may indicate that here to bypass the “Comments to the Author” section, enter your conflict of interest statement in the “Confidential to Editor” section, and submit your "Accept" recommendation.

Reviewer #1: All comments have been addressed

Reviewer #2: All comments have been addressed

2. Is the manuscript technically sound, and do the data support the conclusions?

Reviewer #1: Yes

Reviewer #2: Yes

3. Has the statistical analysis been performed appropriately and rigorously? 

Reviewer #1: Yes

Reviewer #2: Yes

4. Have the authors made all data underlying the findings in their manuscript fully available?

Reviewer #1: Yes

Reviewer #2: Yes

5. Is the manuscript presented in an intelligible fashion and written in standard English?

Reviewer #1: Yes

Reviewer #2: Yes

6. Review Comments to the Author

Reviewer #1: The manuscript has improved form original submission and now looks better. This will be of great implications in future for the field of rapid diagnostics for malarial parasties. This will revolutionise the field in not only diagnostics but also will impact the early treatment.

Reviewer #2: (No Response)

7. PLOS authors have the option to publish the peer review history of their article (what does this mean?). If published, this will include your full peer review and any attached files.

Reviewer #1: **Yes: **RBS Kushwah

Reviewer #2: No

---

## [Editor Report · Acceptance letter]

25 Jan 2024

PONE-D-23-19957R1 

PLOS ONE

Dear Dr. Sikulu-Lord, 

I'm pleased to inform you that your manuscript has been deemed suitable for publication in PLOS ONE. Congratulations! Your manuscript is now being handed over to our production team.

Kind regards, 

on behalf of

Dr. Yash Gupta 

Academic Editor

PLOS ONE